# Effects of Controlled Ozone Exposure on Circulating microRNAs and Vascular and Coagulation Biomarkers: A Mediation Analysis

**DOI:** 10.3390/ncrna9040043

**Published:** 2023-08-01

**Authors:** Hao Chen, Syed Masood, Ana G. Rappold, David Diaz-Sanchez, James M. Samet, Haiyan Tong

**Affiliations:** 1Oak Ridge Institute for Science and Education, Oak Ridge, TN 37830, USA; hchen255@gmail.com; 2Curriculum in Toxicology and Environmental Medicine, University of North Carolina at Chapel Hill, Chapel Hill, NC 27599, USA; masood.syed@epa.gov; 3Public Health and Integrated Toxicology Division, Center for Public Health and Environmental Assessment, U.S. Environmental Protection Agency, 104 Mason Farm Rd, Chapel Hill, NC 27514, USA; rappold.ana@epa.gov (A.G.R.); diaz-sanchez.david@epa.gov (D.D.-S.); samet.james@epa.gov (J.M.S.)

**Keywords:** microRNA, ozone, vascular biomarker, mediation, *miR-19a-3p*, TNFα

## Abstract

Exposure to ozone (O_3_) is associated with adverse respiratory and cardiovascular outcomes. Alterations in circulating microRNAs (miRNAs) may contribute to the adverse vascular effects of O_3_ exposure through inter-cellular communication resulting in post-transcriptional regulation of messenger RNAs by miRNAs. In this study, we investigated whether O_3_ exposure induces alterations in circulating miRNAs that can mediate effects on downstream vascular and coagulation biomarkers. Twenty-three healthy male adults were exposed on successive days to filtered air and 300 ppb O_3_ for 2 h. Circulating miRNA and protein biomarkers were quantified after each exposure session. The data were subjected to mixed-effects model and mediation analyses for the statistical analyses. The results showed that the expression level of multiple circulating miRNAs (e.g., *miR-19a-3p*, *miR-34a-5p*) was significantly associated with O_3_ exposure. Pathway analysis showed that these miRNAs were predictive of changing levels of downstream biomarkers [e.g., D-dimer, C-reactive protein, tumor necrosis factor α (TNFα)]. Mediation analysis showed that *miR-19a-3p* may be a significant mediator of O_3_-exposure-induced changes in blood TNFα levels [0.08 (0.01, 0.15), *p* = 0.02]. In conclusion, this preliminary study showed that O_3_ exposure of healthy male adults resulted in changes in circulating miRNAs, some of which may mediate vascular effects of O_3_ exposure.

## 1. Introduction

Tropospheric ozone (O_3_) is a ubiquitous, highly reactive oxidant pollutant produced by photochemical reactions between nitrogen oxides and volatile organic compounds in the presence of ultraviolet light [1]. Ambient O_3_ levels are regulated by the U.S. Environmental Protection Agency under the Clean Air Act, with the current National Ambient Air Quality Standard (NAAQS) set at 0.07 ppm [1]. Despite significant air quality improvement in recent decades in the U.S., more than 120 million Americans currently live in regions that exceed the NAAQS for O_3_ [2]. Human exposure to O_3_ is associated with respiratory and cardiovascular health impacts [3]. Acute inhalational exposure to O_3_ in clinical studies induces dose-dependent decrements in spirometric parameters, including forced vital capacity (FVC) and forced expiratory volume in the first second (FEV_1_) [4,5]. Controlled O_3_ exposure also induces a transient neutrophilic influx in the airways, elevations in markers of vascular inflammation, as well as changes in markers of fibrinolysis and autonomic control of heart rate and cardiac repolarization [6]. Furthermore, epidemiological studies have revealed positive associations between ambient O_3_ exposure and cardiovascular morbidity and mortality [7,8].

Mechanistically, O_3_ is believed to induce health effects through stimulation of intraepithelial nociceptive nerves, oxidative stress, and inflammation [9]. Inhaled O_3_ quickly reacts with airway surface lining fluid to produce lipid hydroperoxides and oxidizes cellular membranes, the products of which may contribute to dynamic respiratory and cardiovascular responses [10,11]. One such response is transient polymorphonuclear neutrophilic infiltration of the airways and the release of cytokines and chemokines including interleukin 1 (IL-1), IL-6, and tumor necrosis factor alpha (TNFα) into the lung tissue [4,6,12]. However, the mechanisms underlying O_3_-induced cardiovascular effects remain unclear.

While the adverse cardiovascular effects of O_3_ inhalation may be explained by systemic inflammation and oxidative stress regulated at a transcriptional level, it may also be controlled by epigenetic regulatory mechanisms, such as those involving microRNAs (miRNA) [13,14]. miRNAs are short (about 22 nucleotides in length), single-stranded, noncoding RNAs that are vital in the post-transcriptional regulation of gene expression in cells [13,15]. After synthesis, miRNAs are secreted into the circulation, often contained in extracellular vesicles (EV) that have a role in intercellular communication between respiratory cells exposed to air pollutants such as O_3_ and those located in cardiovascular or neurological systems [13,14,16]. miRNAs can regulate post-transcriptional gene expression by inhibiting or destabilizing the translation of mRNA into protein, potentially leading to pathophysiological changes [13]. Thus, it is possible that acute exposure to O_3_ causes adverse changes in vascular and coagulation biomarkers through altered profiles of circulating miRNAs.

Exposure to ambient air pollutants, including fine particulate matter (PM_2.5_), coarse particulate matter (PM_10_), O_3_, and nitrogen oxides (NOx), has been previously associated with altered expression of circulating miRNAs, including *miR-146a-5p*, *miR-150-5p*, *miR-155-5p*, *miR-21-5p*, and *miR-25-3p* [17,18,19,20]. While several studies have investigated miRNAs’ mediational role in the cardiovascular pathophysiology induced by air pollution exposure [19,21,22,23], most of these studies have focused on PM, and only a few have specifically investigated effects of O_3_ exposure. In fact, we recently reported that human exposure to ambient O_3_ induces changes in plasma C-reactive protein (CRP) and total cholesterol levels, possibly through altered expression of circulating *miR-26a-5p* [24]. Thus, more human exposure studies are needed to establish whether circulating miRNAs constitute a viable mechanistic intermediate for O_3_-induced cardiovascular effects.

Mediation analysis is a statistical technique that attempts to quantitatively assess the relative impact of individual pathways and mechanisms through which an exposure affects an outcome [25]. By employing this method, we set out to determine whether certain miRNAs play a significant role in the etiology of the cardiovascular effect of O_3_ exposure. Mono- and poly-unsaturated fatty acids have anti-oxidant and anti-inflammatory properties. We recently reported that the cardiopulmonary effects of acute ozone exposure are modulated by dietary supplementation of fish oil or olive oil in a cohort of healthy volunteers [26]. We further found that acute exposure to 300 ppb O_3_-induced decrements of pulmonary function and elevation in systolic blood pressure were attenuated by dietary supplementation with these beneficial oils [26]. In the present study, we analyzed human plasma samples from this randomized chamber exposure study to specifically investigate whether changes in specific circulating miRNAs are significantly associated with acute human exposure to O_3_ and further examined their possible mediational role in the induction of vascular markers of inflammation. We additionally investigated whether fish oil or olive oil supplementation modulates the mediational pathways of O_3_–miRNA–mRNA/protein biomarkers. The results of this study may offer insights into the molecular mechanisms through which O_3_ exposure induces cardiovascular effects and identify miRNAs that may have utility as biomarkers of the response to O_3_ inhalation.

## 2. Methods

### 2.1. Study Participants

This study is a part of a randomized controlled trial “OMEGOZ” (ClinicalTrials.gov, NCT03395119). Detailed information of study participants and design has been described previously [26]. Briefly, healthy participants having no history of cardiovascular disease, pulmonary disease, cancer, or other diseases, not taking dietary supplements or medications such as β-adrenergic receptor blockers or anti-inflammatory drugs, were enrolled. Written informed consent was given by all participants prior to enrollment. This protocol was reviewed and approved by the Institutional Review Board of the University of North Carolina at Chapel Hill and the U.S. Environmental Protection Agency Human Subjects Review Office.

### 2.2. Study Design

As described previously in the main study [26], eligible participants were randomly assigned into three groups and received no supplements (control, CTL), 3 g of fish oil (FO) daily (Pharmavite, LLC, San Fernando, CA, USA), or 3 g of olive oil (OO) (Arista Industries, Inc., Wilton, CT, USA) daily for 4 weeks. Dietary and medication restrictions were applied to all participants during the study. At the end of the dietary supplementation, participants were exposed for 2 h to filtered air on the first day and to O_3_ (mean concentration 300 ± 30 ppb) on the second day while exercising intermittently to a targeted ventilation (VE) rate at 20 L/min/m^2^ on an ergometer every other 15 min. While the O_3_ concentration of 300 ppb employed in the present study was higher than that of the NAAQS (70 ppb), it is attained episodically in the U.S. and China [27,28]. This concentration of O_3_ also falls in the range that has been previously shown to effectively induce acute pulmonary responses in human chamber studies [4,5,6].

To conduct an exploratory analysis of miRNAs’ mediation role and limit the impacts of sex, this study only assessed O_3_ effects on miRNA and protein biomarkers of 23 male participants. Venous blood samples collected approximately 1 h post filtered air and O_3_ exposure were included in the analysis.

### 2.3. Biomarker Measurement

#### 2.3.1. miRNA Profiling

Sodium citrate plasma samples were stored at −80 °C until assayed. The FirePlex^®^ cardiology panel (ABCAM, Cambridge, MA, USA) was employed to measure the plasma levels of 65 miRNAs which have been associated with cardiovascular health. The method has been described previously [24]. Briefly, 40 μL plasma sample was mixed with Digest Buffer and Protease Mix to a final volume of 80 μL and then incubated at 60 °C for 45 min. A reaction mixture (25 μL of the prepared sample + 35 μL FirePlex Particles + 25 μL of hybridization buffer) was incubated at 37 °C for 60 min in a 96-well plate. After rinsing to remove unbound RNA, 75 μL of labeling buffer was added to each well and incubated for 60 min at room temperature. Adapted-modified miRNAs were eluted using 95 °C water and collected for PCR amplification using a PCR master mix. PCR product was then hybridized with hybridization buffer at 37 °C for 30 min. Particles were then rinsed and scanned on an EMD Millipore Guava 8HT flow cytometer. Raw output was background subtracted, normalized using the geometric mean of three normalizer miRNAs (*miR-15b-5p*, *miR-17-5p*, and *miR-93-5p*).

#### 2.3.2. Protein Marker Measurement

Commercially available ELISA kits were used to quantify plasma levels of interleukin 1β (IL-1β), IL-6, IL-8, tumor necrosis factor alpha (TNFα), CRP, soluble vascular cell adhesion molecule 1 (sVCAM1), soluble intercellular adhesion molecule 1 (sICAM1), serum amyloid A (SAA), E-selectin, D-dimer, and von Willebrand factor (vWF) (MesoScale Diagnostics, Gaithersburg, MD, USA). All assays were carried out according to manufacturer’s instructions.

### 2.4. Data Analysis

#### 2.4.1. Identifying Differentially Expressed Circulating miRNA and Protein Biomarkers

All miRNA and protein data were log2 transformed to approximate normal distribution of residuals. To assess changes in biological endpoints between O_3_ exposure and dietary supplementation, we used a mixed-effects model with a participant-specific random intercept and the Tukey’s tests adjusted for pair-wise comparisons. SAS 9.4 software (Cary, NC, USA) was used for statistical analysis. A preliminary analysis suggested that there was limited modifying effects of dietary supplementation on miRNAs expression in response to O_3_ exposure (Supplemental Appendix A). Due to the small sample size in each group, we did not differentiate the dietary groups when conducting the mediation analysis in this study. Rather, we pooled data from all three dietary groups to increase the sample size and identify O_3_-exposure associated miRNAs while including dietary supplementation as a covariate. We used the statistical results of the Type III Tests of Fixed Effects to estimate the overall effects of O_3_ on the miRNAs [29]. Statistical results of *p* < 0.05 were considered significant.

#### 2.4.2. Functional Annotation between miRNA and Protein Biomarkers

We conducted an in silico analysis using the Ingenuity Pathway Analysis (IPA, Ingenuity Systems^®^, Redwood City, CA, USA) to identify mRNA targets of the statistically significant miRNAs; the miRNA–mRNA relationships were based on experimental results indexed in the established database. Because this was a preliminary study, we did not seek to validate the links between miRNA and mRNA. The analyzed miRNAs were those significantly associated with O_3_ exposure identified in the previous step, and mRNAs were for those protein biomarkers that have been measured in this study.

#### 2.4.3. Mediation Analysis

Mediation analysis was employed to assess the statistical significance of the O_3_–miRNA–protein biomarker matches. The methods have been detailed previously [24]. In brief, we estimated the statistical significance of the indirect effects of miRNAs on the association between exposure to O_3_ and protein biomarkers. This approach decomposes the total estimated effects of air pollutants on the biomarker expression into an estimate of the direct effect and an estimate of the indirect effect of air pollutants on the biomarker expression mediated through miRNA. To allow for a comparable analysis, we standardized the data of exposure (O_3_ exposure), mediator (miRNAs), and outcomes (protein biomarkers) by subtracting the mean and dividing by the standard deviation. Two models have been fitted using the generalized linear mixed model (Figure 1): (1) model “X → M” was to assess the association between O_3_ exposure (X) and miRNAs (M) and acquire coefficient of X as “a”; (2) model “X + M → Y” was to assess the link between miRNAs and protein biomarkers (Y) while also considering O_3_ exposure (X), acquiring coefficient of M as “b”. Sobel test was conducted to evaluate if the indirect effect (a × b) was statistically significant. The results of *p* < 0.05 are considered as “statistically significant”.

## 3. Results

### 3.1. Participant Characteristics

Venous blood samples obtained from 23 male participants (6, 7, and 10 in CTL, FO, and OO groups, respectively) were included in the analysis (Table 1). The average age of the participants was 26 years old, and the BMI was 25.3. There was no statistical difference in the mean age and BMI among the three groups. There was a statistically significant difference in the number of participants who self-identified as being members of a specific race/ethnicity between the three groups (*p* = 0.04).

### 3.2. Effects of Dietary Supplementation and O_3_ Exposure on Circulating miRNAs

Descriptive statistics for the 65 miRNAs that were analyzed in the blood samples are presented in Appendix A. Among these miRNAs, dietary supplementation with FO or OO did not significantly alter the expression of the assessed miRNAs following filtered air or O_3_ exposure with the exception of the level of *miR-34a-5p* expression which was significantly elevated by FO supplementation compared with that of the CTL after exposure to filtered air. However, we found that the expression levels for seven miRNAs were significantly altered by O_3_ exposure in at least one of the dietary groups. Figure 2 presents log-transformed expression data of miRNAs significantly affected by O_3_ exposure. For example, the expression of circulating *miR-122-5p* was significantly elevated after O_3_ exposure compared with that of filtered air exposure [1.78 (1.56, 2.01) vs. 2.02 (1.67, 2.37), *p* = 0.007] in the CTL diet group. Compared with those of filtered air exposure, changes in circulating miRNA levels after O_3_ exposure were significantly altered for *miR-125b-5p* [−0.54 (−1.04, −0.04) vs. −1.27 (−1.77, −0.78), *p* = 0.04] and *miR-144-5p* [−1.88 (−3.88, 0.12) vs. −0.72 (−1.22, −0.22), *p* = 0.03] in the FO group, *miR-155-5p* in the CTL group [−0.64 (−1.15, −0.13) vs. −0.13 (−0.25, −0.01), *p* = 0.047], *miR-19a-3p* in the OO group [1.16 (1.08, 1.24) vs. 1.01 (0.91, 1.10), *p* = 0.008], *miR-342-3p* in the FO group [1.05 (0.89, 1.21) vs. 1.24 (1.11, 1.36), *p* = 0.02], and *miR-34a-5p* in the CTL group [−0.54 (−1.19, 0.11) vs. 0.11 (−0.44, 0.67), *p* = 0.01].

### 3.3. miRNAs Associated with O_3_ Exposure

As shown above, dietary supplementation did not have a significant effect on miRNA expression after filtered air exposure. O_3_ exposure was associated with changes in the expression of several miRNAs in at least one of the dietary groups. However, the sample size was relatively small in each dietary group and insufficient to conduct a meaningful mediation analysis. In order to examine miRNAs’ role in mediating the effects between O_3_ exposure and vascular biomarkers, we considered the overall effects of O_3_ on specific miRNA levels by pooling data from all three dietary groups together, while controlling dietary supplementation status as a covariate. We employed type III statistics in SAS, which examined for overall fixed effects of parameters for O_3_ and dietary supplementation (Appendix A). Because the effect of dietary supplementation on miRNA parameters were relatively small, the role of dietary supplementation is not further considered in this study. Among the 65 miRNAs, type III statistics showed that O_3_ exposure was associated with the expression of circulating *miR-122-5p* (F = 7.09, *p* = 0.015), *miR-144-5p* (F = 6.39, *p* = 0.045), *miR-192-5p* (F = 4.37, *p* = 0.0496), *miR-194-5p* (F = 7.83, *p* = 0.011), *miR-199a-5p* (F = 7.14, *p* = 0.015), *miR-19a-3p* (F = 5.47, *p* = 0.030), and *miR-34a-5p* (F = 4.77, *p* = 0.044) (Table 2).

### 3.4. Proteins That Are Predicted to Be Downstream Biomarkers of miRNAs

After identifying miRNAs associated with O_3_ exposure, we conducted a pathway analysis to search for potential matches between protein targets measured in the study and specific miRNAs. In this analysis, we examined the circulating levels of proteins associated with inflammation, coagulation, and vascular reactivity, including CRP, D-dimer, E-selectin, IL-6, IL-8, IL-1β, SAA, TNFα, sICAM1, sVCAM1, and vWF. The comparisons on theses biomarkers among different dietary groups and O_3_ exposures were reported previously [26]. We also observed elevated blood levels of IL-6 and decreased E-selectin levels post O_3_ exposure in at least one of the dietary groups among the male participants (Appendix A). Although not all biomarkers were significantly affected by O_3_ exposure, we presumed that mediational effects of miRNAs could still occur even if the independent variable (i.e., O_3_ exposure) was not significantly associated with a dependent variable (i.e., protein biomarkers) [30]. Based on this assumption, we further identified possible “O_3_–miRNA–protein biomarker” matches using the IPA application. As shown in Table 2, four out of the seven O_3_ exposure-associated miRNAs were predicted to regulate downstream mRNA/protein targets. Specifically, expression change in *miR-194-5p* was predicted to regulate the changes in circulating levels of D-dimer. Expression changes in circulating *miR-199a-5p* and *miR-19a-3p* were predicted to regulate the changes in circulating levels of CRP and TNFα, respectively, following O_3_ exposure. In addition, *miR-34a-5p* expression levels were predicted to regulate circulating CRP and sVCAM1 (Figure 3).

### 3.5. Mediation of miRNAs between O_3_ Exposure and Changes in Vascular and Coagulation Biomarkers

Next, we conducted a mediation analysis to determine whether the predicted “O_3_–miRNA–protein” matches were statistically significant, potentially validating the mediational role of miRNAs in the effects of O_3_ exposure on their downstream biomarkers. This analysis investigated the indirect effects of M (miRNAs) between X (O_3_ exposure) and Y (protein biomarkers) (Figure 1). Among the five predicted “O_3_–miRNA–protein” matches, “O_3_–*miR-19a-3p*–TNFα” showed a statistically significant indirect effect of O_3_ exposure on TNFα levels through changes in circulating *miR-19a-3p* (Table 3). Specifically, the expression changes in circulating *miR-19a-3p* significantly down-regulated approximately 67.1% of O_3_ exposure-induced effects on TNFα [indirect effects, 0.08 (0.01, 0.15), *p* = 0.02]. We did not observe any other significant indirect effects of miRNAs between O_3_ exposure and protein biomarkers. In addition, we investigated possible moderating effects of dietary supplementation on the significant mediational model (O_3_–*miR-19a-3p*–TNFα) as suggested in Figure 1. However, we did not find a link to suggest that dietary supplementation with FO or OO significantly alters the mediational pathways: X → M [interaction between X (O_3_) and W (dietary supplementation): F = 1.91, *p* = 0.17] or X + M → Y [interaction between X (O_3_) exposure and W (dietary supplementation): F = 1.22, *p* = 0.30) (Appendix A).

## 4. Discussion

Exposure to tropospheric O_3_ has been associated with increased morbidity and mortality; however, the mechanisms underlying such health impacts remain to be elucidated. Circulating miRNAs could serve as a possible mediator from O_3_-exposed pulmonary cells to those in the extrapulmonary system. In this human study, we investigated whether exposure to O_3_ altered blood levels of coagulation and vascular inflammation biomarkers through changes in circulating miRNAs among healthy male participants.

It has been shown previously that exposure to O_3_ is associated with altered expression of certain miRNAs in humans. For instance, exposure to ambient O_3_ among healthy adults was significantly associated with changes in circulating miRNAs, including *let-7e-5p*, *miR-125-5p*, and *miR-26a-5p* [24]. Another study also demonstrated significant changes in blood exosomal *miR-150-5p* and *miR-155-5p* in coronary artery disease patients exposed to short-term ambient O_3_ [17]. In a controlled chamber exposure study, exposure to 400 ppb O_3_ for 2 h induced altered expression of several miRNAs (e.g., *miR-143*, *miR-145*, *miR-199a*, *miR-222*, and *miR-25*) in induced sputum samples of healthy volunteers, suggesting the involvement of innate immune responses [31]. In the present study, among the 65 circulating miRNAs we investigated, 7 were associated with O_3_ exposure, including *miR-122-5p*, *miR-125b-5p*, *miR-144-5p*, *miR-155-5p*, *miR-19a-3p*, *miR-342-3p*, and *miR-34a-5p*. Therefore, our findings are consistent with previously published studies showing that O_3_ exposure leads to alterations in the expression of specific miRNAs.

The function of miRNA is to regulate the expression of target genes/proteins. The RNA-inducing silencing complex (RISC), which is a cytoplasmic multi-protein complex formed with single stranded RNA fragments including miRNAs, is able to regulate the translation of target mRNA through a post-initiation step [32]. miRNAs released into the extracellular space can reach different tissues and organs, thereby relaying biological information through regulation of mRNA expression to promote cell-to-cell communication that is essential in the cardiovascular effects of lung exposure to air pollutants [16]. O_3_ exposure—associated miRNAs identified in the present study—have been linked to cardiovascular pathophysiology. *let-7e-5p* expression is correlated with MAP kinase activation that involves CASP3 and TGFBR1, serving as a biomarker for ischemic stroke [33]. *miR-122-5p* is a key regulator of the SIRT6-elabela-ACE2 signaling pathway that is involved in angiotensin II-mediated apoptosis and oxidative stress in vascular endothelial cells [34]. Circulating *miR-150-5p*, *miR-342-3p*, and *miR-34a-5p* are also being investigated as biomarkers of acute heart failure [35] and vascular inflammation associated with vascular diseases [36,37,38]. Therefore, these altered expression levels of miRNAs may implicate multiple molecular mechanisms involved in cardiovascular disease induced by O_3_ exposure.

The altered circulating miRNAs in this study are believed to be involved in biological signaling pathways including IL-6 signaling, atherosclerosis signaling, and acute phase response signaling, implicating vascular inflammation and injury. Furthermore, the mediation results suggested that circulating *miR-19a-3p* may significantly relay the effects of exposure to O_3_ on the downstream inflammation marker TNFα. Elevated expression of *miR-19a-3p* has been reported in blood samples in patients with cardiovascular diseases such as acute heart failure [39]. In vitro studies have shown that overexpression of *miR-19a-3p* may promote vascular inflammation [40] and neuroinflammation [41]. It is possible that acute O_3_ exposure may stimulate a defense mechanism that downregulated expression of pro-inflammatory miRNA *miR-19a-3p*, which could inhibit pro-inflammatory cytokines such as TNFα in circulation.

Dietary supplementation with unsaturated fatty acids from fish oil or olive oil has been associated with cardiopulmonary protection against exposure to ambient air pollution [8,26,42,43]. Fish oil is rich in omega-3 polyunsaturated fatty acids such as eicosapentaenoic acid (EPA) and docosahexaenoic acid (DHA) that may mitigate adverse effects of air pollution exposure through antioxidant and anti-inflammatory properties [44]. Olive oil is rich in oleic acid, a monounsaturated fatty acid that has also shown health benefits [45]. In the present study, we did not focus on the impacts of dietary supplementation on miRNAs and their derived mediation analysis due to the small sample size in each dietary group. However, functional studies have suggested that omega-3 PUFA may modify the biological signaling pathways of several miRNAs. For example, consistent with our finding that FO treatment was associated with elevated expression of circulating *miR-34a-5p*, EPA and DHA may modulate the activation of p53/*miR-34a-5p*/Bcl-12 axis in myeloma cells [46]. A previous study reported that dietary intake of omega-3 PUFA may significantly modify both direct and indirect effects between air pollutant exposure and cardiovascular biomarkers through *miR-26a-5p* [24]. Future studies with larger sample size are warranted to examine the role of dietary interventions on the miRNA-mediated air pollution health effects.

The results of this study suggest a possible mediational role of miRNAs in O_3_-exposure-induced vascular inflammation and that certain miRNAs may be biological biomarkers in the mechanistic investigation of O_3_ induced health effects. However, there are a few limitations of this study. First, this study was a preliminary investigation with a relatively small sample size and was limited to young healthy male participants. Using a small sample size may be prone to type II statistical errors and the study may not be generalizable to a broader population. Second, because we only assessed blood biomarkers at one timepoint, it is possible that a more sensitive timepoint in which to observe potentially larger effects of O_3_ exposure on miRNAs was missed. Third, only protein levels of inflammation, coagulation, and vascular injury biomarkers were assessed, thus limiting our interpretation of the mRNA levels of these markers, which are direct regulating targets of miRNAs. In addition, besides miRNAs, we did not consider other possible mediational pathways through systemic inflammation and oxidative stress. We also cannot assume causal relationships of O_3_–miRNA–protein pathways, especially when miRNA and protein markers were assessed at the same time, raising the need for future functional studies of the specific pathway. Finally, statistical results were not adjusted for multiple testing, increasing the chance of both type I and type II errors. Despite these limitations, this exploratory study suggests that miRNAs may be able to mediate the acute effects of O_3_ exposure on vascular inflammation.

## 5. Conclusions

To our knowledge, this is the first study investigating whether circulating miRNAs may mediate the effects of inhalational exposure to O_3_ on coagulation and vascular inflammation biomarkers among healthy participants in a controlled chamber exposure trial. We have found that several circulating miRNAs (e.g., *miR-122-5p*, *miR-144-5p*, *miR-19a-3p*, *miR-34a-5p*) were significantly altered by O_3_ exposure. Among those, *miR-19a-3p* is a possible intermediator between O_3_ exposure and change in blood TNFα levels, consistent with miRNA–mRNA matches identified in an existing database. These findings offer preliminary evidence of the role of circulating miRNAs as a biological mechanism of vascular inflammation induced by O_3_ exposure.

## Figures and Tables

**Figure 1 ncrna-09-00043-f001:**
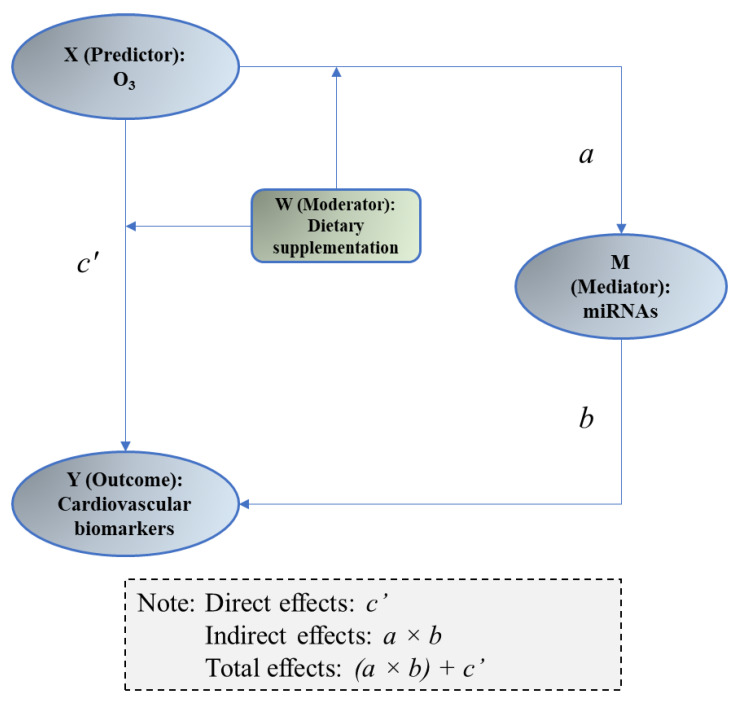
Schematic showing the mediation model employed in this study. Letter “a” indicates the coefficient of X derived from the statistical analysis for “X → M”, “b” indicates the coefficient of M in the model “M → Y”, and “c” indicates the coefficient of X in the model “X  +  M → Y”.

**Figure 2 ncrna-09-00043-f002:**
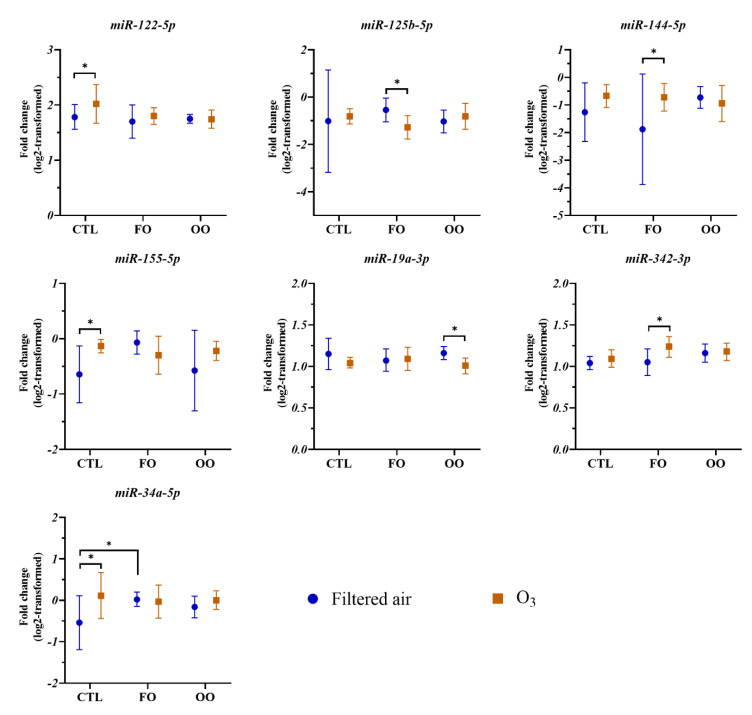
Changes in circulating miRNAs for participants in control (CTL), fish oil (FO), and olive oil (OO) groups immediately after exposure to filtered air or 300 ppb O_3_. Shown are log2-transformed fold change values in *miR-122-5p*, *miR-125b-5p*, *miR-144-5p*, *miR-155-5p*, *miR-19a-3p*, *miR-342-3p*, and *miR-34a-5p*. Bars with whiskers indicate means with their respective 95% confidence interval. * *p* < 0.05 as “significant” between two conditions.

**Figure 3 ncrna-09-00043-f003:**
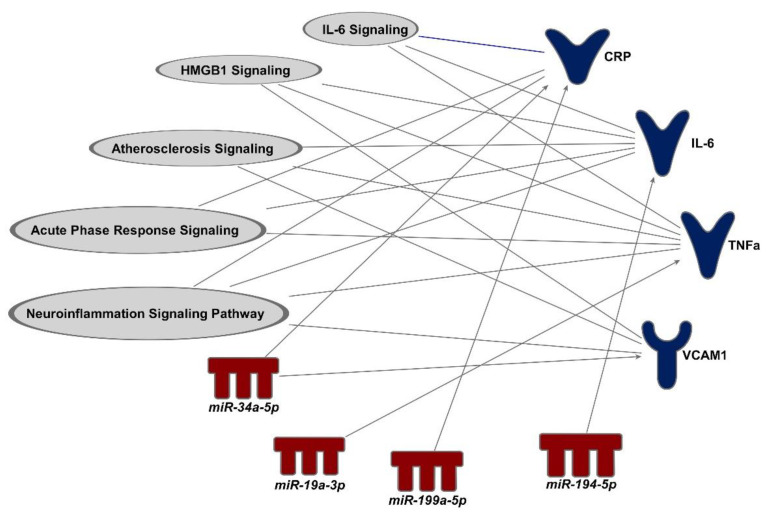
Pathway analysis showing the link between O_3_ exposure-induced miRNAs and vascular proteins in the plasma and their related biological pathways. Four miRNAs (in red) that were associated with O_3_ exposure and their matched mRNA/protein targets (in dark blue) and their associated downstream inflammatory signaling pathways (in gray) were identified by the Ingenuity Pathway Analysis software.

**Table 1 ncrna-09-00043-t001:** Participant demographics.

Characteristics	CTL (n = 6)	FO (n = 7)	OO (n = 10)	All (n = 23)
Age (years)	23.5 (3.7)	27.4 (4.7)	26.5 (3.0)	26.0 (3.9)
BMI (kg/m^2^)	24.9 (3.6)	25.8 (3.8)	25.1 (2.7)	25.3 (3.2)
Omega-3 index (%)	4.0 (0.2)	6.1 (1.2)	4.0 (0.4)	4.6 (1.2)
Race/ethnicity (no. of participants)				
African-American	0	2	0	2
Asian	1	1	0	2
Caucasian	3	4	10	17
Hispanic	2	0	0	2

BMI: body mass index; CTL: control; FO: fish oil; OO: olive oil.

**Table 2 ncrna-09-00043-t002:** O_3_ exposure—associated miRNAs and their predicted downstream cardiovascular targets.

miRNAs	*F* Value	*p*	Predicted Targets
*miR-122-5p*	7.09	0.015 *	-
*miR-144-5p*	6.39	0.0448 *	-
*miR-192-5p*	4.37	0.0496 *	-
*miR-194-5p*	7.83	0.0111 *	D-dimer
*miR-199a-5p*	7.14	0.0146 *	CRP
*miR-19a-3p*	5.47	0.0299 *	TNFα
*miR-34a-5p*	4.77	0.0441 *	CRP, sVCAM1

O_3_ exposure associated miRNAs were identified using a mixed-effects model with subject as random effect and adjusting for covariates including dietary supplementation and the interaction product of O_3_ exposure and dietary supplementation. * *p* < 0.05 O_3_ exposure was a “significant” factor for changes in the circulating miRNA. Measured protein biomarkers were predicted and matched with the identified miRNAs using the Ingenuity Pathway Analysis. CRP: c-reactive protein, IL-6: interleukin 6, IL-8: interleukin 8, sVCAM1: soluble vascular cell adhesion molecule 1, TNFα: tumor necrosis factor α.

**Table 3 ncrna-09-00043-t003:** Mediation effects of miRNAs on the association between exposure to ozone and biomarkers.

miRNAs (M)	Biomarker (Y)	X → M (a)	M → Y (b)	Direct Effects (c′)	Indirect Effects (a × b)		
					Coefficient (95%CI)	Proportion (%)	*p*
*miR-194-5p*	D-dimer	0.45 (−0.2, 1.1)	**0.16 (0.09, 0.24)**	**−0.17 (−0.29, −0.05)**	0.07 (−0.04, 0.19)	−76.78	0.20
*miR-199a-5p*	CRP	−0.38 (−1.07, 0.31)	0.02 (−0.05, 0.09)	**0.34 (0.22, 0.46)**	−0.01 (−0.04, 0.02)	−2.17	0.64
*miR-19a-3p*	TNFα	**−1.03 (−1.81, −0.25)**	**−0.08 (−0.11, −0.05)**	**−0.21 (−0.28, −0.13)**	**0.08 (0.01, 0.15)**	−67.09	0.02 *
*miR-34a-5p*	CRP	0.32 (−0.53, 1.18)	**0.22 (0.16, 0.27)**	**0.27 (0.16, 0.37)**	0.07 (−0.11, 0.25)	20.79	0.46
*miR-34a-5p*	sVCAM1	0.32 (−0.53, 1.18)	**0.05 (0.02, 0.08)**	0.05 (−0.01, 0.11)	0.02 (−0.03, 0.06)	25.63	0.47

The letters X, M, and Y denote independent variable (X for O_3_ exposure), mediator (M for miRNAs), and dependent variable (Y for cardiovascular biomarkers). The letters a, b, and c′ denote coefficients of X, M, and X in their respective models shown in Figure 1. Data were presented as the coefficients with 95% confidence interval (95%CI). Bold font indicates that the coefficient was statistically significant (*p* < 0.05). The statistical significance of indirect effects (a × b) was assessed using the Sobel Test. * *p* < 0.05 indicates that the indirect effects were statistically significant. Proportion of indirect effects in the total effects (a × b + c′) was calculated. CRP: c-reactive protein, TNFα: tumor necrosis factor α, sVCAM1: soluble vascular cell adhesion molecule 1.

## Data Availability

The data presented in the current study will be made available in ScienceHub (https://catalog.data.gov/dataset/epa-sciencehub).

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
