# Peer review of "Effects of Controlled Ozone Exposure on Circulating microRNAs and Vascular and Coagulation Biomarkers: A Mediation Analysis"

_ncrna, 2023, doi:10.3390/ncrna9040043_

Round 1

Reviewer 1 Report (New Reviewer)

https://www.ncbi.nlm.nih.gov/pmc/articles/PMC7306136/ 

similar study was conducted 

Author Response

  1. https://www.ncbi.nlm.nih.gov/pmc/articles/PMC7306136/, similar study was conducted.

Response: We appreciate the reviewer noticing our previous research. Although these studies are superficially similar, this manuscript examines the relationship between higher concentration of ozone exposure and expression of circulating miRNAs in far greater detail. For example, the current study was based on ozone chamber exposure where causal inference is stronger than the panel study design utilized in the previous research. In addition, the current study also employed a mediation model to investigate the role of miRNAs between ozone exposure and cardiovascular biomarkers.

Reviewer 2 Report (New Reviewer)

It is a good study having biological importance 

Author Response

  1. It is a good study having biological importance.

Response: We appreciate reviewer’s recognition of our work.

Reviewer 3 Report (New Reviewer)

Overall, this is a great paper with a very good logical flow of writing structure. The authors presented a novel insight on the impacts of ozone on a variety of circulating miRNAs and downstream mRNAs/proteins. It would be even better if the authors could have dug into more details and conducted more experiments on specific miRNAs or proteins to uncover their interactions, but as the authored declared, this is a preliminary investigation on this subject.  However, I would suggest being more conservative on drawing the conclusions as there was no direct evidence presented in this study on the correlations of the miRNAs and specific cardiovascular markers. Some comments and questions to be clarified are listed below. 

1. In average, what level of O3 do people usually get exposed to given that the NAAQS regulation was 0.07ppm? How was the 300 ppb treatment determined? Were there any evidence showing the efficacy of this level of stimulation? 

2. IPA is a huge database for pathway analyses. Did you input any experimental data to predict your target miRNAs or did you solely rely on the established connections in the IPA? Please elaborate a bit more on how you narrow down to the 65 miRNAs.  

3. What was the rationale for pre-supplementing fish oil and olive oil? It seems like there were some references mentioned in the discussion section, but is there a way to provide some level of information in the introduction so that the readers won’t feel they came out all of a sudden? How did you narrow down to only comparing these two dietary supplementations? Were there any presumable effects on treating these two supplements? 

4. In supplemental Table 1, miR-19a-3p seems to be down-regulated by O3 in the OO group. However, in Table 3, it seems like the author suggested that miR-19a-3p was up-regulated by O3 (X->M) and then the increased miR-19a-3p down-regulated TNFa (M->Y) causing an indirect effect. Please explain.

5. Line 368: mRNA/protein

6. Line 366 to 382 are simply repeating the results. No new information or discussions were included.

Author Response

  1. Overall, this is a great paper with a very good logical flow of writing structure. The authors presented a novel insight on the impacts of ozone on a variety of circulating miRNAs and downstream mRNAs/proteins. It would be even better if the authors could have dug into more details and conducted more experiments on specific miRNAs or proteins to uncover their interactions, but as the authored declared, this is a preliminary investigation on this subject. However, I would suggest being more conservative on drawing the conclusions as there was no direct evidence presented in this study on the correlations of the miRNAs and specific cardiovascular markers. Some comments and questions to be clarified are listed below. 

Response: We appreciate the reviewer’s compliments. As the reviewer suggested, we have revised the language to be more conservative in the conclusion regarding the relationship between miRNAs and specific cardiovascular markers, in Lines 33-35 in the abstract and Lines 357-361 in the Conclusion section.

  1. In average, what level of O3 do people usually get exposed to given that the NAAQS regulation was 0.07ppm? How was the 300 ppb treatment determined? Were there any evidence showing the efficacy of this level of stimulation? 

Response: The exposure concentration of ozone at 300 ppb is higher than the NAAQS, but this level is reached during photochemical smog episodes in Southern California (Environmental Protection Indicators for California, https://oehha.ca.gov/media/downloads/risk-assessment/document/epicupdate2004a.pdf) and in Beijing, China (Wang 2006, https://agupubs.onlinelibrary.wiley.com/doi/full/10.1029/2006GL027689). The main clinical study was a mechanistic study that investigated the cardiopulmonary effects of ozone exposure and the efficacy of fish oil and olive oil intervention. The concentration of 300 ppb has also been employed in multiple chamber exposure studies to elicit significant short-term decrements in lung function and an elevation of neutrophil counts in the airway. The rationale of employing this ozone concentration has been added in the “Methods” section (Line 121-124).

  1. IPA is a huge database for pathway analyses. Did you input any experimental data to predict your target miRNAs or did you solely rely on the established connections in the IPA? Please elaborate a bit more on how you narrow down to the 65 miRNAs.  

Response: As the reviewer indicated, we mainly relied on the established connections in the IPA database for the links between miRNA and mRNA/protein targets. Future experiments will further validate the identified links between ozone-miRNA-mRNA/protein markers. We have elaborated this point in the Methods section (Line 168-169). We used the commercial FirePlex® cardiology panel (ABCAM, Cambridge, MA) of 65 miRNAs that differentially regulated in normal or abnormal cardiovascular health.

  1. What was the rationale for pre-supplementing fish oil and olive oil? It seems like there were some references mentioned in the discussion section, but is there a way to provide some level of information in the introduction so that the readers won’t feel they came out all of a sudden? How did you narrow down to only comparing these two dietary supplementations? Were there any presumable effects on treating these two supplements? 

Response: We thank the reviewer for the suggestions. The main aim was to investigate the impacts of ozone on circulating miRNAs and their downstream mRNA/protein targets. Secondly, we also wanted to investigate whether dietary supplementation of fish oil or olive oil moderate such associations. Because of the small sample size, we only touched on fish oil or olive oil supplementation briefly. Now, we have added more introduction on the inclusion of fish oil and olive oil in the present study (Line 87-92).

  1. In supplemental Table 1, miR-19a-3p seems to be down-regulated by O3 in the OO group. However, in Table 3, it seems like the author suggested that miR-19a-3p was up-regulated by O3 (X->M) and then the increased miR-19a-3p down-regulated TNFa (M->Y) causing an indirect effect. Please explain.

Response: We apologize and appreciate the reviewer for pointing out this error in our original analysis.  We have rechecked the analysis and found that when we were running the mixed effects model for the coefficients for ozone treatment, the software SAS set “1 (ozone exposure)” as the reference group, instead of “0 (filtered air group)”. Therefore, we obtained coefficients for O3 (X->M) as upregulation which counteracted the downregulation as we observe for the raw data. We have revised the analysis setting “0 (filtered air group)” as the reference group. The numerical values of the coefficients changed direction for the X as the outcome of the adjustment. However, the statistical results for the indirect effects did not change the outcome for the mediation tests. Overall, we have updated Table 3 as well as the main text to show the directional changes of the coefficients of the predictor X.

  1. Line 368: mRNA/protein

Response: Thank you for the proofreading.

  1. Line 366 to 382 are simply repeating the results. No new information or discussions were included.

Response: We now simplified this section to avoid repetition and merged this section into the next one where function of miR-19a-3p was discussed (Line 307-316).

Reviewer 4 Report (New Reviewer)

In the manuscript by Chen et al., the authors investigated the association between ozone exposure and circulating miRNAs, and drew the conclusion that the expression level of multiple circulating miRNAs was associated with ozone exposure. However, the evidence shown by the authors is weak in supporting their conclusions. The authors themselves admit that the sample size was small and more importantly, they only performed this one-time experiment. Under this design, the data could be completely random and artifact. And their results in Figure 2 did not show significant differences, although some showed slight statistical significance, regardless of the long bars. Under the basis of such results, the analyses are unreliable, and the conclusions can be misleading. So, I recommend the authors collect more samples and gain more reliable results. Also, the authors should detect other RNAs that are of relatively stable expression level as a control to show that the miRNA change is not a result of global change of all circulating RNAs. Some further experimental validations will be needed for their conclusions as well. One more point is that the authors should explain why they set the three groups (CTL, FO, OO) before describing their results, like in the introduction.

Author Response

  1. In the manuscript by Chen et al., the authors investigated the association between ozone exposure and circulating miRNAs, and drew the conclusion that the expression level of multiple circulating miRNAs was associated with ozone exposure. However, the evidence shown by the authors is weak in supporting their conclusions. The authors themselves admit that the sample size was small and more importantly, they only performed this one-time experiment. Under this design, the data could be completely random and artifact. And their results in Figure 2 did not show significant differences, although some showed slight statistical significance, regardless of the long bars. Under the basis of such results, the analyses are unreliable, and the conclusions can be misleading. So, I recommend the authors collect more samples and gain more reliable results. Also, the authors should detect other RNAs that are of relatively stable expression level as a control to show that the miRNA change is not a result of global change of all circulating RNAs. Some further experimental validations will be needed for their conclusions as well. One more point is that the authors should explain why they set the three groups (CTL, FO, OO) before describing their results, like in the introduction.

Response: We appreciate the reviewer’s constructive comments. We fully agree with the reviewer that the relatively small sample size may render the findings of this study prone to both type I and type II statistical errors. However, due to the difficulty of running a clinical trial and human chamber exposure studies, it was not possible to collect hundreds or thousands of samples. But the advantage of human chamber exposure studies is that we have a stronger confidence of causal inference between exposure and outcomes. As the reviewer suggested, we will expand the sample size and conduct both in vivo and in vitro experiments in our future studies to validate the current findings. We noted the confusion of not introducing the three groups in the introduction and now have added a few sentences in the last paragraph of the Introduction section (Line 87-92).

Reviewer 5 Report (New Reviewer)

The authors have tried to understand the effect of controlled ozone exposure on circulating microRNAs and vascular as well as coagulation biomarkers.

-Could authors please address what is the novel outcome of the study?

-As authors also agreeing, the sample count used in the study are little less for such kind of study, could you please address that?

-Is study not biased if only 65 microRNAs and few selective markers are used to study the effect?

-So authors are concluding that diet has no effect?

-Why there was no validation performed for the outcomes?

-How the exposure of 1 hour was selected for air? Is it justifiable?

Author Response

  1. The authors have tried to understand the effect of controlled ozone exposure on circulating microRNAs and vascular as well as coagulation biomarkers. Could authors please address what is the novel outcome of the study?

Response: The novel outcome of the current study is that we were able to determine several circulating miRNAs that were significantly impacted by acute exposure to ozone in a chamber exposure study. We also found that miR-19a-3p may be a significant mediator of the effects of ozone exposure on circulating TNFa.

  1. As authors also agreeing, the sample count used in the study are little less for such kind of study, could you please address that?

Response: As addressed to the comment of Reviewer #4, we agree with the reviewer that the relatively small sample size may render the findings of this study prone to both type I and type II statistical errors. However, due to the difficulty of running a clinical trial and human chamber exposure studies, it was not possible to collect hundreds or thousands of samples. But the advantage of the human chamber exposure study is that we have a stronger confidence of causal inference between exposure and outcomes.

  1. Is study not biased if only 65 microRNAs and few selective markers are used to study the effect?

Response: The aim of this study was to investigate circulating miRNAs, which are believed to be related to cardiovascular health and were from a pre-selected cardiology panel, that may be affected by acute ozone exposure and whether the significantly changed miRNAs mediate ozone effects on the downstream vascular inflammation markers. We agree that the selected miRNAs and inflammation markers may not represent the full spectrum of biological molecules involved in cardiovascular health, but this preliminary investigation provides evidence for more comprehensive studies in the future. Future studies will employ a more global screening technique to include more miRNAs and mRNAs.

  1. So authors are concluding that diet has no effect?

Response: This preliminary study did not find any significant moderating impacts of either fish oil or olive oil on the ozone-miRNA-protein link. However, it does not exclude the possibility with a larger sample size in the future.

  1. Why there was no validation performed for the outcomes?

Response: This preliminary study is mainly based on the clinical trial samples and data. Validation experiments on the identified ozone-miRNA-protein pathway will be warranted in our future studies.

  1. How the exposure of 1 hour was selected for air? Is it justifiable?

Response: As the methods section indicated, both filtered air and ozone exposure lasted for 2 hours. (Line 118-119)

Round 2

Reviewer 1 Report (New Reviewer)

Effects of controlled ozone exposure on circulating microRNAs and vascular and coagulation biomarkers: a mediation analysis : more novel approach is needed

Author Response

We appreciate the reviewer’s time and effort in evaluating our manuscript. In this study, we applied a cardiology panel of miRNAs and a mediation model to assess the role of miRNAs in the association between exposure to ozone and cardiovascular biomarkers in human participants exposed to ozone. From an epidemiological perspective, this analysis identifies the role of mediators in the association between predictors and outcomes but, as we have acknowledged in the manuscript, experiments to validate the mediators’ role is lacking are beyond the scope of this exploratory project and will require additional research to investigate.

Reviewer 3 Report (New Reviewer)

The authors have satisfactorily addressed all the comments.

Author Response

Thank you!

Reviewer 4 Report (New Reviewer)

The authors haven't addressed my previous concerns about sample size and the reliability of the conclusions.

Author Response

We appreciate the reviewer’s time and effort in evaluating our manuscript. Once again, we fully agree with the reviewer’s concern about the reliability of the conclusions due to the small sample size. Ideally, increasing the number of participants would be beneficial; however, enrollment in this controlled exposure study has ended and it is not possible to do so.  Nonetheless, as we have pointed out in the Discussion, while a small sample size may render results prone to statistical errors, we believe the findings of our present analyses to be sound and sufficiently important to merit publication at this time. We would add that, the FirePlex Cardiology panel employed a method to choose three relatively stable miRNAs as internal control for the analysis of the miRNA expression, considering the variability of miRNA expression among studies.  We wholeheartedly agree with the reviewer that the results of this study should be followed by future controlled clinical exposure studies, as well as mechanistic in vitro experiments.

This manuscript is a resubmission of an earlier submission. The following is a list of the peer review reports and author responses from that submission.

Round 1

Reviewer 1 Report

The manuscript titled "Effects of controlled ozone exposure on circulating microRNAs and vascular and coagulation biomarkers: a mediation analysis" was evaluated and my comments are below.

1- The logic of doing the project is not well explained in the introduction.

2- Figures and tables are not well designed.

3- Rewrite the part of measuring molecular biomarkers more clearly.

4- Please mention take-home-massage for biological applications of your results in the discussion.

5- The manuscript should be checked in terms of structure and grammar.

Author Response

Response to Reviewers’ Comments

Reviewer #1

The manuscript titled "Effects of controlled ozone exposure on circulating microRNAs and vascular and coagulation biomarkers: a mediation analysis" was evaluated, and my comments are below.

  • The logic of doing the project is not well explained in the introduction.

Response: We thank the reviewer for the comment. We agree that the rationale of this study needs to be explained more clearly. We now have revised the last paragraph of the introduction to state that that this study’s aim is to evaluate miRNA’s role in O3-induced vascular effects and identify miRNAs that may serve as biomarkers of the effects of ozone (Line 85-92).

2- Figures and tables are not well designed.

Response: We appreciate the reviewer’s comment. We have now revised Figures 1 and 3 to show a schematic of the mediation model and IPA pathway results, respectively. We have also checked for errors in the tables and corrected as appropriate. 

3- Rewrite the part of measuring molecular biomarkers more clearly.

Response: We thank the reviewer for the suggestion. We have carefully revised the methods part of measuring molecular biomarkers, particularly adding more details on the miRNA measurement. As for the protein biomarkers, standard ELISA kits were used, and the assays were carried out according to manufacturer's instructions. Line 120-132. 

4- Please mention take-home-massage for biological applications of your results in the discussion.

Response: We appreciate the reviewer’s suggestion. The take-home-message for the biological applications of our results is the identification of potential biomarkers of the adverse response to inhalational exposure to air pollutant such as ozone.  (Line 337-339)

5- The manuscript should be checked in terms of structure and grammar.

Response: We have invited native English speakers in our group to double check the structure and grammar. The text has been revised as necessary.

Reviewer 2 Report

In this article, the authors studied the impact of O3 exposure on the expression of miRNAs in healthy male adults.

It will be interesting to study, the impact of O3 in females and the miRNA regulation. This study could have been more impactful if the authors could have included both males and females in the current study itself (also increase the number of study participants).

Authors identified the miRNA target by IPA, provide the IPA signaling network figures,

Provide observation made here, in the form of a signaling network.

Validate the miRNA targets at mRNA levels and Protein by western. Authors validated using specific KIT,  it will be good to see the target gene expression levels at the mRNA  and visualize by western blot as well.

The author needs to discuss their finding succinctly relevant to vascular inflammation.

Author Response

Reviewer #2

In this article, the authors studied the impact of O3 exposure on the expression of miRNAs in healthy male adults.

  1. It will be interesting to study, the impact of O3 in females and the miRNA regulation. This study could have been more impactful if the authors could have included both males and females in the current study itself (also increase the number of study participants).

Response: We wholeheartedly agree with the reviewer on this point that the study would be more impactful with a larger representative sample size that includes both sexes. With this limited sample selection, we have observed possible mediation of miR-19a-3p on the association between O3 exposure and TNFa. This finding will provide a basis for a follow up study that will focus on a larger sample size that include both sexes.  Line 340-342.

  1. Authors identified the miRNA target by IPA, provide the IPA signaling network figures,

Provide observation made here, in the form of a signaling network.

Response: We thank the reviewer for the suggestion. An IPA signaling network has now been generated and included in the main text shown as Figure 3.

  1. Validate the miRNA targets at mRNA levels and Protein by western. Authors validated using specific KIT, it will be good to see the target gene expression levels at the mRNA and visualize by western blot as well.

Response: This is a great suggestion. Validating the identified miRNA targets would offer more direct evidence of the O3 – miRNA – mRNA/protein links. In a separate study, we have also investigated the associations between O3, miRNA, and vascular inflammation markers using an epidemiological study design (Chen, 2022. DOI: 10.1016/j.ecoenv.2022.113604). The validation of the O3 – miR-19a-3p – TNFa link will be thoroughly investigated in a separate project using both in vitro and in vivo models and benchwork methodologies such as PCR and Western Blotting. The lack of functional validation of the identified pathway has been mentioned as a study limitation and future research direction in the discussion. Line 348-351.

  1. The author needs to discuss their finding succinctly relevant to vascular inflammation.

Response: We thank the reviewer for the suggestion. We have revised the discussion section to limit redundant content in terms of vascular inflammation related to our findings. Line 286-294. Line 297-299.

Round 2

Reviewer 2 Report

The authors need to provide experimental evidence, which I suggested earlier, that will make this paper more interesting and reach the audience better.